# Nuclear Morphofunctional Organization and Epigenetic Characteristics in Somatic Cells of *T. infestans* (Klug, 1834)

**DOI:** 10.3390/pathogens12081030

**Published:** 2023-08-11

**Authors:** Maria Luiza S. Mello

**Affiliations:** Department of Structural and Functional Biology, Institute of Biology, University of Campinas, Campinas 13083-862, SP, Brazil; mlsmello@unicamp.br

**Keywords:** insect vector, somatic polyploidy, chromocenters, Malpighian tubules, epigenetics, valproic acid, valproate

## Abstract

*Triatoma infestans* (Klug) is an insect recognized as not only an important vector of South American trypanosomiasis (Chagas disease) but also a model of specific cellular morphofunctional organization and epigenetic characteristics. The purpose of the present review is to highlight certain cellular processes that are particularly unveiled in *T. infestans*, such as the following: (1) somatic polyploidy involving nuclear and cell fusions that generate giant nuclei; (2) diversification of nuclear phenotypes in the Malpighian tubules during insect development; (3) heterochromatin compartmentalization into large bodies with specific spatial distribution and presumed mobility in the cell nuclei; (4) chromatin remodeling and co-occurrence of necrosis and apoptosis in the Malpighian tubules under stress conditions; (5) epigenetic markers; and (6) response of heterochromatin to valproic acid, an epidrug that inhibits histone deacetylases and induces DNA demethylation in other cell systems. These cellular processes and epigenetic characteristics emphasize the role of *T. infestans* as an attractive model for cellular research. A limitation of these studies is the availability of insect supply by accredited insectaries. For studies that require the injection of drugs, the operator’s dexterity to perform insect manipulation is necessary, especially if young nymphs are used. For studies involving in vitro cultivation of insect organs, the culture medium should be carefully selected to avoid inconsistent results.

## 1. Introduction

*Triatoma infestans* (Klug), the most important vector of Chagas disease, is a hemipteran insect belonging to the Triatominae subfamily of the Reduviidae family and one of the 84 species of the Triatomini tribe. This species developed through the Andean and non-Andean evolutionary lineages with different geographic distributions in South America, genomic size, and main repetitive DNA [1]. Both lineages dispersed from Bolivian highlands, from where they had originated. Forty-two satellite DNA families are present in varying amounts in both lineages [2].

The Andean lineage was primarily dispersed in the high-altitude valleys of Bolivia and Peru and in lower altitudes of southern Peru; the non-Andean lineage occupied lowland regions in Argentina, Brazil, Chile, Paraguay, Uruguay, and Bolivia and certain higher altitude regions in Argentina until chemical interventions for insect control were undertaken in the past decades, which drastically reduced these groups [2]. Although these lineages differ in nuclear DNA content per haploid genome (1487 Gbp, non-Andean lineage; 1936 Gbp, Andean lineage) and in the amounts of satellite DNA families, they have the same number of holokinetic chromosomes (2n = 20 + XY or XX) [2].

In addition to the importance of *T. infestans* as a vector for *Trypanosoma cruzi* infection, it has been revealed as a model for studies on unusual cellular processes involving somatic polyploidy, nuclear and cell fusion, presence of large heterochromatin bodies (chromocenters), diversity of nuclear phenotypes, cell death, epigenetic markers, and chromatin response to histone deacetylase inhibition. This unprecedented review briefly describes these phenomena in *T. infestans* specimens collected in Brazil and subsequently reared in insectary facilities at 30 °C and 80% relative humidity. The somatic cell peculiarities reported here point out *T. infestans* as an attractive model for cell biology investigations.

## 2. Somatic Polyploidy

Insects of most orders develop somatic polyploidy, in which DNA endoreplication steps advance and permit nuclear and cellular volume enlargement and organ growth without intervention of mitosis [3]. This phenomenon occurs mainly during the larval stages in insects undergoing complete metamorphosis (holometabola), or in nymphal instars of insects undergoing simple metamorphosis (hemimetabola) [3]. Hemipterans develop through hemimetaboly. Somatic polyploidy involving an integral doubling of the genome does not occur in all insect groups; specific genomic regions, as in the case of the heterochromatin, may replicate differentially [4].

Somatic polyploidy may be considered an adaptive process through which the cells save the physiological time and protein synthesis that would be required if a mitotic spindle and nuclear membrane were fabricated as in the case of ordinary somatic division [3]. Some authors have hypothesized that it is a strategy for increasing gene copy numbers, which would facilitate high rates of biosynthesis and metabolism [4]. In healthy mammal tissues, somatic polyploidy has been considered important for cells to function under stress, protecting their viability and permitting their adaptation under new environmental conditions [5,6].

The DNA endoreplication steps that occur in the nuclei of the Malpighian tubule cells of fully-nourished *T. infestans* specimens are arrested at the end of the fifth nymphal instar, when Feulgen-DNA 32 C and 64 C optimal ploidy degrees are attained per nucleus compared with the Feulgen-DNA content of nuclei in the first nymphal stages and in spermatids [7,8]. Feulgen-DNA classes or ploidy degrees that are identified as 1 C, 2 C, 4 C, and so on, are associated with the DNA content of haploid, diploid, tetraploid, and so on, nuclei of the same species [9]. However, in fasted insects, the ordinary polyploidization pattern detected in the Malpighian tubules is surpassed because nuclear and cellular fusions occur even with low incidence in specimens reared in the laboratory [10]. Then, ploidy degrees shift to considerably higher values, as demonstrated by DNA cytophotometric quantification, visual detection of the presence of giant nuclei (Figure 1a), or counting of the total number of nuclei/organ that decrease significantly [11]. Giant nuclei may attain a ploidy degree as high as 1024 C [11]. Consequently, the total number of nuclei may show a 40% reduction [12]. Additionally, nuclear and cell fusions have been detected in BTC-32 cells cultivated from embryonic tissues of *T. infestans* at the London School of Hygiene and Tropical Medicine [13]. The presence of giant nuclei is not restricted to *T. infestans*; it has also been reported in the fat body cells of *Rhodnius prolixus*, another species of the Reduviidae family [14].

Initially, nuclear fusion is facilitated by the binucleate composition of the somatic cells [15]. When fasting periods are prolonged, cell fusion and additional nuclear fusions occur in the Malpighian tubules. Cell fusion may be favored by the presence of the viral particles that infect these cells, especially when the insects are fed avian blood under laboratory conditions [15,16]. Cell fusion has been suggested to represent an attempt at cell survival that opposes cell death [11,14]; however, both phenomena may coexist in the same organ [10].

## 3. Heterochromatin and Nuclear Phenotypes

Condensed chromatin aggregates comprising constitutive heterochromatin (chromocenters) are present in *T. infestans* cells from the first nymphal instar [7]. These bodies increase in size through DNA endoreplication steps that do not always accompany the DNA doubling process in the euchromatin (DNA underreplication), representing a constitutive heterochromatin characteristic [7]. Other indicators of the constitutive heterochromatin nature of these condensed chromatin bodies are as follows: the absence of RNA transcriptional activity in most regions of these bodies, as detected using radioautography; abundance of repetitive DNA; and differential response to the Feulgen reaction compared with the characteristics of euchromatin [7,17,18].

The chromocenters of *T. infestans* are usually composed of copies of the large A, B, and C autosomes plus the X and Y sex chromosomes [19,20]. One additional autosome pair has been reported as part of this heterochromatin type under specific insect rearing conditions [21]. Additionally, polymorphism and variable amount of heterochromatin have been reported for *T. infestans* [22,23,24].

From the first to the third nymphal instar, only one chromocenter is present in the Malpighian tubules of *T. infestans*; from this stage onward, a portion of the cells exhibit more than one chromocenter (Figure 1b–d) [7,8]. The process of formation of multi-chromocenters apparently involves budding or declustering from the single chromocenter body [25] and does not lead to the micronuclei production that results in cell death as reported for other cell systems [26]. The nuclear phenotype diversification and the size attained by chromocenters in the Malpighian tubules of nymphs and adults have not been identified in other species of the Reduviidae family studied till date [27]. These unexplained phenomena could not be shown to be associated with diversified functions exerted by the Malpighian tubules, which are excretory organs eliminating the byproducts of a blood diet that does not change during the insect lifetime.

Notably, free DNA phosphates showed differential availability for the electrostatic binding of toluidine blue molecules under Mg^2+^ competitive staining conditions, when the single-chromocentered nuclei were compared with nuclei bearing multiple chromocenters [28]. This finding revealed that the dye-binding sites in the DNA of chromocenters in the multi-chromocentered nuclei are more exposed and easier to block by Mg^2+^ ions than those in the DNA of chromocenters in single-chromocentered nuclei probably owing to differences in the packing states of the DNA-protein complexes involved [29].

*T. infestans* nuclei constitute an attractive model for studying the 3D position of chromocenters because of the large volume of these bodies inside the nuclei. Using confocal microscopy and quantitative measurement of the 3D position of the chromocenters in the Malpighian tubule cells of *T. infestans*, the proximity of these bodies to the nuclear periphery was demonstrated during the insect lifetime [25]. The spatial distribution of chromocenters in relation to the nuclear periphery was confirmed using electron microscopy [25].

The proximity of chromocenters to the nuclear periphery has been detected in the cells of several other organisms [29,30,31,32], including species of the Reduviidae family which exhibit very small chromocenters, as in the case of *Panstrongylus megistus* [33]; however, the reason for this phenomenon remains unclear [34]. The concept that proximity of a chromocenter to the nuclear periphery is associated with overall gene silencing [35,36] does not fully apply to *T. infestans*, because 18 active S rDNA sites were detected in the chromocenter surface including the region facing the nuclear periphery, according to videos obtained after 3D fluorescence in situ hybridization [25]. In *T. infestans* cells, the nucleolus was found to encircle chromocenters (Figure 2a,b) [37]. Additionally, the hypothesis that heterochromatin acts as a gene-silencing compartment has been questioned [30]. Variability relative to the small distance between the edge of the chromocenters and the nuclear periphery raises the suspicion of the mobility of the heterochromatin body with respect to the nuclear envelope [25].

Unraveling of the chromocenter heterochromatin has been observed in a small number of nuclei in the Malpighian tubules of *T. infestans* under rearing conditions in the laboratory (Figure 1d) [10]. However, the frequency of this phenotype was observed to be significantly increased in fasted nymphs and in specimens subjected to heat shock [10,11]. This phenomenon, which affects the heterochromatin in other cell systems and under different physiological conditions has been hypothesized to represent an attempt to activate silent genes possibly present in this chromatin compartment and achieve cell survival [38,39].

Jagannathan et al. [26,40], through studies in *Drosophila* and mouse cells, proposed that satellite DNA and their binding proteins may physically integrate the entire chromosome complement into a chromocenter network thus ensuring encapsulation of all chromosomes within a single cell. Notably, images of the epithelial cell nuclei of *T. infestans* suggest confluence of chromosomes that do not properly integrate the chromocenters into this heterochromatin cluster (Figure 3a–c). This hypothesis can also be suggested when examining published electron microscopy images of *T. infestans* cell nuclei [37].

## 4. Response to Stress Agents

*T. infestans* is resistant to long fasting periods. The Malpighian tubules of fifth instar nymphs starved for 6.5 months not only exhibit an increased frequency of giant nuclei, while the frequency of single- and multi-chromocentered nuclei decreases, but also an increased frequency of the nuclear phenotype in which chromocenter decondensation is evident [11]. In addition to the nuclear phenotypes usually detected in the Malpighian tubules of fully-nourished specimens of *T. infestans*, and the giant nuclei resulting from cell fusion, nuclei with morphological characteristics representing cell death or exhibiting chromocenter decondensation may be detected under stress conditions (Figure 3d,e).

Heat and cold shocks have been reported to elicit different responses with respect to survival of the specimens and frequency of nuclear phenotypes in the Malpighian tubules compared with fasting conditions. Seven days after a one-hour heat shock treatment at 40 °C, the frequency of single- and multi-chromocentered nuclei decreased while the frequency of the nuclear phenotype with chromocenter decondensation increased, giant nuclei were few, and nuclear images showed cell death types (apoptosis and especially necrosis) [10]. Thirty days after the one-hour heat shock treatment in fasted specimens, the frequency of giant nuclei increased, while that of nuclei exhibiting chromocenter decondensation was low [10]. Subsequently, giant nuclei affected by apoptosis [42], as well as necrosis substituting for apoptosis, were also observed [10]. Whether the presence of giant nuclei and nuclei with chromocenter unpacking can be considered cellular survival attempts, they were not effective enough to cope with the metabolic events induced by the 1-h heat shock treatment, because the cell death effects surpassed the survival attempts.

When insects were subjected to heat shock at 40 °C for 12 h, the frequency of the Malpighian tubule nuclei containing unpacked chromocenters reached up to 33% relative to the total nuclei immediately after the shock and reduced abruptly to 3.35% and 0.39% after 3 and 30 days, respectively, following the shock [43]. Conversely, the frequency of the giant nuclei increased up to 8.5% relative to the total nuclei in the 30-day period. Although the frequency of cell death images did not surpass 3.5% of the cell nuclei in the Malpighian tubules from insects 30 days after the 12-h heat shock, the total number of nuclei of approximately 12,000 in the controls reduced to 4500–6000 after this shock [43].

In the Malpighian tubules of *T. infestans*, giant nuclei have been reported to be few in nymphs examined 30 days after a 0 °C shock for 1 h and unobservable after a cold shock for 12 h. Chromocenter unpacking is rare under both cold shock periods. While apoptosis is also rare, necrosis frequency may reach approximately 7% relative to the total cell nuclei 30 days after the 12-h cold shock [43].

Regarding survival of the *T. infestans* specimens subjected to temperature shocks, their resistance to heat and cold shock for 1 h at 40 °C and 0 °C has been demonstrated to some extent, despite the above-mentioned cellular effects visualized in their Malpighian tubules. Heat and cold shock conditions did not affect nymphal survival in a 30-day period, although they affected the hormonal balance which controls molting and slightly decreased the survival of male and female adults [44]. However, when heat shock periods at 40 °C were extended to 12 h, survival decreased to 28% in third instar nymphs, 32% in male adults, and 52% in female adults. Nevertheless, the fourth and fifth instar nymphs of *T. infestans* were not found to be affected by this long shock [44].

Fully-nourished or fasted fifth instar specimens of *T. infestans* have shown tolerance to cold shock at 0 °C for up to 12 h even if subjected to a single shock or to sequential shocks at intervals of 8 or 24 h at 30 °C and examined 32 days later [45]. This response to cold shock may have enabled the species to survive low temperatures at their original altitudes in the Andes Mountains before adaptation to Neotropical areas throughout regions of the South America. Such a survival response differs significantly from the less than 10% survival of fifth instar nymphs of *Panstrongylus megistus* after a 12 h shock at 0 °C [45,46]. *P. megistus* is another reduviid species vector of Chagas disease and well disseminated in Brazil. *T. infestans* showed a decreased frequency of apoptosis and necrosis in the Malpighian tubules following sequential cold shocks under full nourishment or 15-day fasting before the initial shock, confirming the tolerance of this insect to cold stress [47].

In a study on the response of *T. infestans* to heavy metal ions, another stress agent, fasted fifth nymphal specimens showed 90% survival up to 24 h after injection of 2 × 10^−5^ M CuSO_4_ and 1.56 × 10^−4^ M CH_3_HgCl; however, after this period, survival abruptly decreased, especially in fully-nourished specimens. In the presence of these heavy metals, necrosis was the most frequent phenotype in the Malpighian tubules, occurring predominantly 2 h after treatments; less than 20% of the single- and multi-chromocentered nuclei exhibited unpacked chromocenters, and very few giant nuclei and apoptotic nuclei were detected [48]. These findings demonstrated the lack of resistance of *T. infestans* to heavy metals.

## 5. Epigenetic Markers: DNA and Histone Methylation

The chromocenters of both single- and multi-chromocentered nuclei of *T. infestans* contain AT-rich, GC-poor DNA, as demonstrated using cytochemistry and immunofluorescence analysis [18]. Cytogenetics, immunolabeling, and molecular biology assays have identified at least four chromosome rearrangements involved in the amplification/dispersion of AT-rich satellite DNA [24]. Fourier transform infrared (FTIR) microspectroscopy confirmed AT-rich DNA in the whole nuclear genome of *T. infestans* [49].

While DNA 5-methylcytosine (5mC) has not been immunocytochemically revealed in chromocenters of the Malpighian tubules of *T. infestans*, it occurs in the euchromatin that is especially concentrated around the chromocenters [18,49]. This absence of 5mC in the heterochromatin is not limited to *T. infestans*; it has also been reported in hemipteran coccids and aphid species [50,51].

Histone epigenetic markers such as acetylated H3K9 (H3K9ac) and H4K8 (H4K8ac) and methylated and di-methylated H3K9 (H3K9me/me2) and H4K20 (H4K20me/me2) have not been detected immunocytochemically in the chromocenters of both single- and multi-chromocentered nuclei of *T. infestans*. However, they have been detected in the euchromatin of these nuclei [52]. These findings are consistent with the differential identification of such markers, except H3K9me2, in the euchromatin compared with that in the heterochromatin in other cell systems [53,54,55,56]. H3K9me2 is generally associated with gene silencing and constitutive heterochromatin formation [53,57]. However, the chromocenters of *T. infestans* do not exhibit positive fluorescence signals in response to anti-H3K9me2 antibody, while euchromatin is positively stained [52]. H4K8ac and H3K9me signals in euchromatin appear concentrated around the chromocenter in single-chromocentered nuclei [52]. This is the region where active 18-S rDNA sites have been demonstrated [25].

The immunofluorescence signals for acetylated H4K16 (H4K16ac) in the images reported for *T. infestans* are not sufficiently clear to discard the presence of this marker in the heterochromatin body [52]. In addition, H4K16ac has been associated with gene transcriptional activities by some authors [58,59] but with transcriptional repression by others [60]. Therefore, the results reported for this marker in *T. infestans* and their implications are inconclusive till date and require in-depth investigation possibly using 3D confocal microscopy and molecular biology assays.

Tri-methylated H3K9 (H3K9me3) marker predominates in the chromocenters of single- and multi-chromocentered nuclei of *T. infestans*; additionally, positive signals are observed in euchromatin granular points (Figure 4a,b) [52]. This epigenetic marker along with methylated H3K27 has been associated with constitutive heterochromatin and inactive enhancers in other cell systems [56,61]. The finding of H3K9me3 signals in euchromatin points may indicate their position in promoters of repressed genes [52]. Tri-methylated H4K20 (H4K20me3), which is considered a constitutive heterochromatin hallmark [55,62], is also revealed in the chromocenters of *T. infestans* (Figure 4c–f).

## 6. Response to Epigenetic Modulators

Valproic acid (VPA) is a short-chain branched fatty acid, and its application extends from beyond its primary and wide use in the treatment of neurological disorders to the modulation of epigenetic markers, including the following: inhibition of histone deacetylases, which favors histone acetylation; changes in the DNA and histone methylation status; direct binding to chromatin components; and chromatin remodeling [56,63,64,65,66,67,68,69,70,71,72,73]. VPA has also been revealed as a promising agent against a series of tumors, owing to its inhibitory effects on cell proliferation and induction of cell cycle arrest and cell death [74,75,76,77].

Insects have been suggested as an alternative model for research on epigenetic effects promoted by VPA, because of their shorter generation intervals, the opportunity for analyzing larger samples, and diminished ethical concerns compared with those for the experimental use of mammals. Therefore, *T. infestans* may be considered an attractive candidate for evaluating pathway mechanisms underlying VPA action [78].

In the presence of VPA, *T. infestans* cells revealed unexpected results [52,78,79]. One-hour following the abdominal injection of a 0.05 mM VPA solution in *T. infestans* nymphs or after a 4-h in vitro culture of the Malpighian tubules of this insect in the presence of 0.05–0.5 mM VPA, unpacking of the chromocenter was demonstrated in a portion of the Malpighian tubule cells (Figure 4g) [52,79]. Higher VPA concentrations lead to co-occurrence of apoptosis and necrosis in the cells of this tissue [79]. Observation of chromocenter unpackaging, which was not accompanied by induction of H3K9 acetylation, differed from the reports for other cell types [68]. An alternative explanation for this finding would be that DNA cytosine demethylation occurred following VPA treatment. However, this hypothesis does not apply to *T. infestans* chromocenters because their DNA is poor or lacking in 5mC [49]. No explanation has been proposed for the induction of the chromocenter decondensation in *T. infestans* cells in the presence of VPA, not involving H3K9 acetylation and DNA demethylation. It certainly requires investigation of other molecules participating in the orchestration of the condensed state of the chromocenters, such as H3K9me3, H4K20me3, the heterochromatin-associated protein (HP1-α) and other non-histone proteins. In other cell systems, H3K9me3 furnishes a binding site to HP1 [80,81]. Regulation of H4K20me3 can occur via an interaction between lysine methyltransferase 5C (KMT5C) and HP1 [55].

## 7. Concluding Remarks

In addition to its relevance as a vector of a disease that affects more than six to seven million people in the Americas, *T. infestans* presents a model for certain cellular peculiarities that may have evolved and contributed to its biological resilience. Most unanswered questions rely on the nature and role of the chromocenters in the Malpighian tubules of this insect. Regardless of whether changes in the number of chromocenters per nucleus are associated with changes in cell fate [32], the disruption of the single chromocenter affecting only a portion of the nuclei in the Malpighian tubules after the third nymphal instar remains to be explained. If chromocenter formation contributes to nuclear stiffening and resistance to small nuclear deformations independent of lamins [82], the declustering of a single chromocenter into a variable number of chromocenters, with no loss of structural and mechanical nuclear integrity, requires interpretation. The hypothetical role of chromocenters in forming a network with chromosomes that do not properly integrate the heterochromatin body needs demonstration of involvement of binding proteins to satellite DNA as reported for mouse and *Drosophila* chromocenters [26,40]. Mobility of the chromocenters inside the cell nuclei, as assumed based on the variability of the distance between the chromocenters and the nuclear periphery, could be supported by in vivo observations. Further studies are also necessary to determine the implication of the unraveling of the chromatin encapsulated into chromocenter bodies under stress conditions or in response to the epidrug VPA, with no association of H3K9 acetylation. These studies include examination and analysis of 3D images representative of other epigenetic markers. Further research on these questions and on establishing a comparative analysis of *T. infestans* cell biology characteristics with other species of the Triatomini tribe and species of the Rhodnini tribe, including *Rhodnius prolixus*, that are also relevant as vectors of Chagas disease, would certainly unveil new and exciting findings that corroborate the use of *T. infestans* as a unique biological model. Particularly, if considering the occurrence of conspicuous heterochromatin bodies, *Panstrongylus megistus* exhibits a small chromocenter contributed by the Y chromosome [83], and *R. prolixus* cell nuclei lack a distinct chromocenter [27]. A limitation of these studies is the availability of insect supply by accredited insectaries. For studies that require injection of drugs, the operator’s dexterity to perform insect manipulation is necessary, especially if young nymphs are used. For studies involving in vitro cultivation of insect organs, the culture medium should be carefully selected to avoid inconsistent results.

## Figures and Tables

**Figure 1 pathogens-12-01030-f001:**
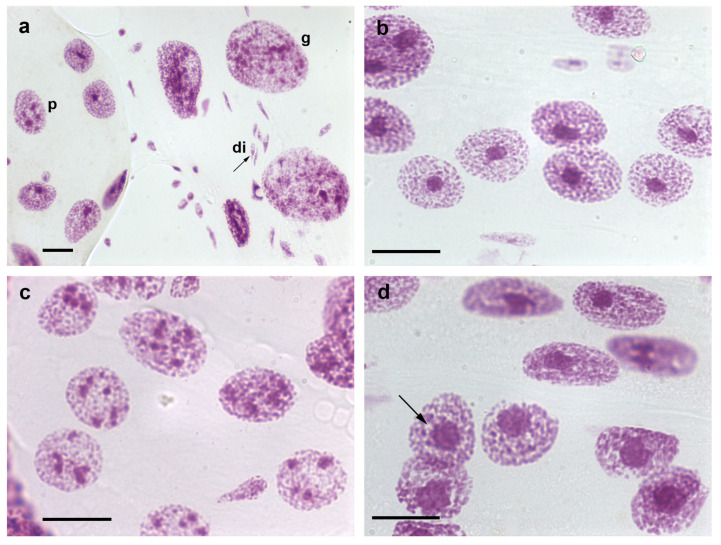
Nuclear phenotypes of Feulgen-stained cells of *T. infestans*. The Feulgen reaction stains DNA specifically [9]. Co-occurrence of diploid fibroblasts (di), and polyploid (p) and giant (g) Malpighian tubule cell nuclei is shown in (**a**). Chromocenters are well evident as conspicuous condensed chromatin bodies in single- (**b**,**d**) and multi-chromocentered nuclei (**c**). A nuclear phenotype showing chromocenter heterochromatin unpacking ((**d**), arrow) can be compared with images of nuclei with a chromocenter usual size ((**b**) or at the up corner of (**d**)). Bars, 20 µm.

**Figure 2 pathogens-12-01030-f002:**
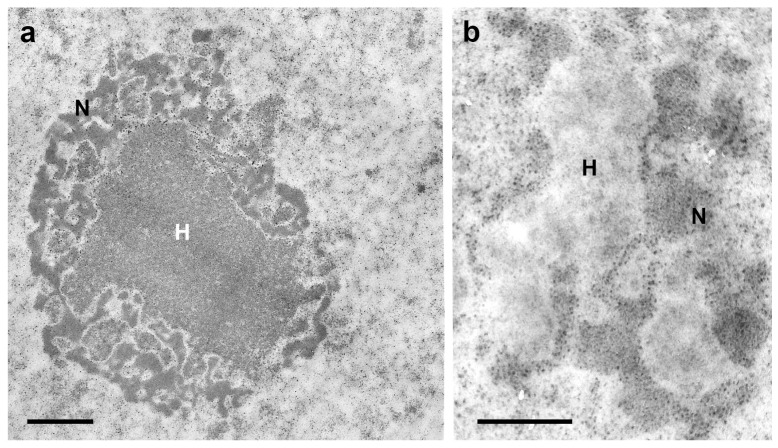
Electron microscopy images of a chromocenter (H)-nucleolus (N) association in Malpighian tubule cell nuclei of fully-nourished fifth instar nymphs of *T. infestans* (**a**,**b**). DNA bleaching in H using Bernard’s EDTA treatment method (**b**) is compared with an untreated control (**a**). This method demonstrates electron dense RNA-rich areas (nucleolar components) [37] that surround the electron lucent DNA-rich heterochromatin zones. (M.L.S. Mello and H. Dolder—unpublished images). Bars, 1 µm (**a**); 0.5 µm (**b**).

**Figure 3 pathogens-12-01030-f003:**
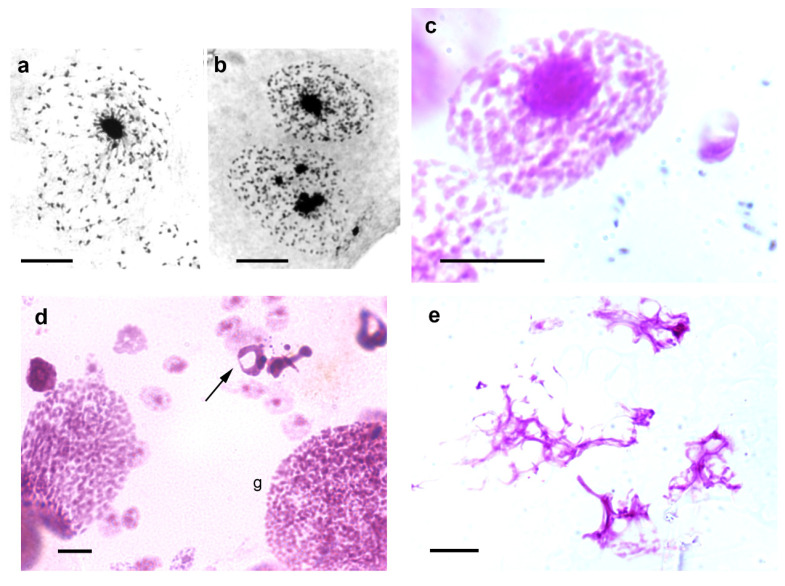
Images suggesting chromosome confluence to chromocenters (**a**–**c**), and cell death images (**d**,**e**) in Malpighian tubules of *T. infestans* nymphs. (**a**,**b**). Toluidine blue-stained cell nuclei. (**c**–**e**). Feulgen-stained cells. Apoptosis ((**d**), arrow) and necrosis (**e**) are evident. Both Toluidine blue and Feulgen stainings indicate DNA-containing nuclear elements [9,41]. g, giant nucleus. Bars, 10 µm (**a**–**c**,**e**); 20 µm (**d**).

**Figure 4 pathogens-12-01030-f004:**
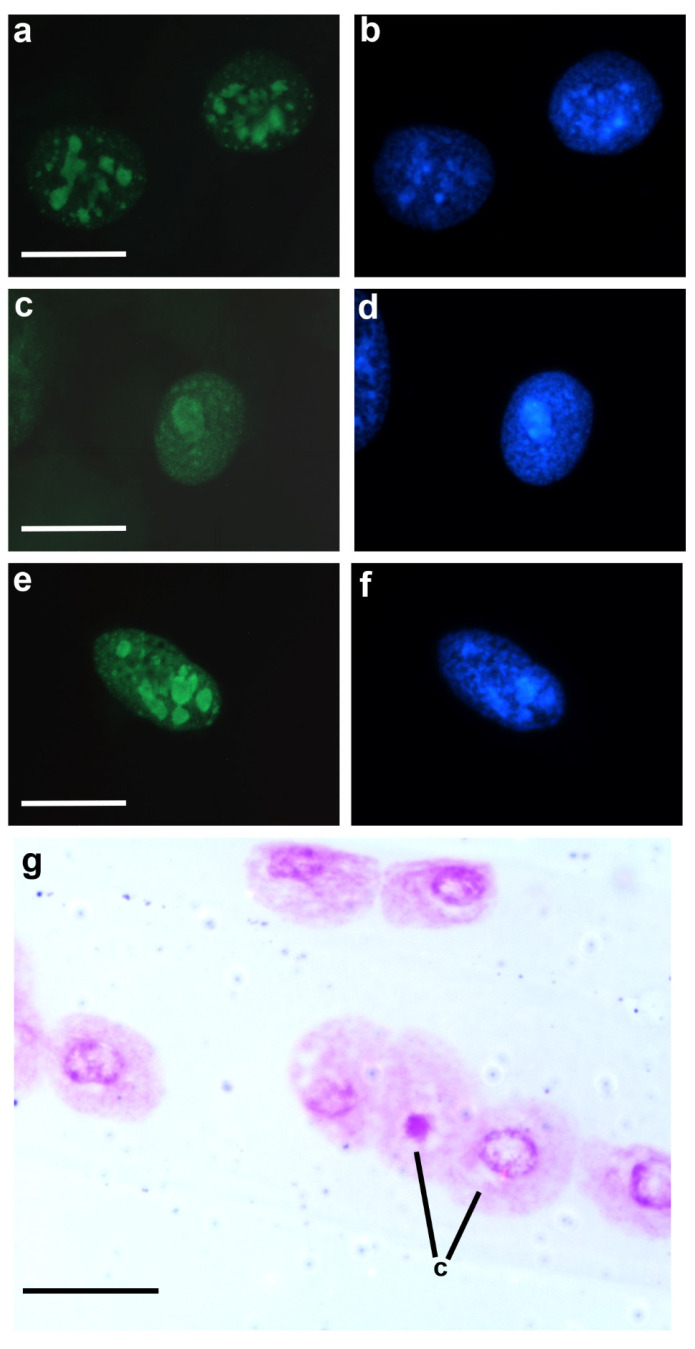
Immunofluorescence signals for H3K9me3 (**a**,**b**) and H4K20me3 (**c**–**f**), and chromocenter heterochromatin unpacking (**g**) in Malpighian tubule cell nuclei of fifth instar nymphs of *T*. *infestans*. Fluorescence intensity of H3K9me3 and H4K20me3 markers is more intense in the chromocenters (**a**,**c**,**e**). Nuclear counterstaining was performed using 4′,6-diamidino-2-phenylindole,dehydrochloride (DAPI) (**b**,**d**,**f**). Chromatin remodeling affecting chromocenters compared with one unaffected chromocenter body (**c**) is detected in Feulgen-stained Malpighian tubule cell nuclei one hour after the insects were injected with a 0.05 mM VPA solution (**g**). The Feulgen reaction stains DNA specifically [9]. Bars, 20 µm.

## Data Availability

All data are found in the mentioned references.

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
