# Peer review of "Nuclear Morphofunctional Organization and Epigenetic Characteristics in Somatic Cells of T. infestans (Klug, 1834)"

_pathogens, 2023, doi:10.3390/pathogens12081030_

Round 1

Reviewer 1 Report

Dear author, I appreciate your submission and the opportunity to review your work. After a careful analysis, I have made some minor modifications to the text to further enrich the effort already put into the revision. It is important to note that these suggestions aim to enhance the content and do not alter the essence of your work.

Title change: In the title, replace "Triatoma infestans (Klug)" with "T. infestans (Klug, 1834) (Hemiptera: Reduviidae: Triatominae)."

Introduction – Could you provide more information about the subfamily, then proceed to discuss the genus Triatoma, and finally delve into the study model, in this case, Triatoma infestans?

One question that arose is whether these images are exclusive for this review and have never been used before? If they haven't, that's excellent. However, if they have been used previously, permission will be required from the journals in which they were published.

In the conclusion, you direct towards the possibility of Triatoma infestans becoming a biological model. I believe you should also include a paragraph highlighting the importance of this research and how it could be applied to other vector species, such as Rhodnius prolixus.

Lastly, congratulations on the effort and dedication demonstrated in crafting this work. I am confident that it will bring significant contributions to the field of study in question.
Best

Author Response

Reviewer #1

Thank you very much for your generous and stimulating comments on my work. Consistent with your suggestions, the title of the article was changed and information on the Triatominae subfamily and the Triatomini tribe was introduced. All figures in the current version of the manuscript are exclusive. Although the images in Fig. 2 in the original version of the manuscript had been reprinted with permission from the Brazilian Journal of Biology, I preferred to substitute new, unpublished figures to the previously used ones. Thank you for calling my attention to this point.

I also included a mention to possible comparisons of cell biology characteristics of T. infestans to other Triatominae species in the Conclusion section.

Thank you again!

Reviewer 2 Report

The current manuscript reviews the literature summarizing the nuclear morphofunctional and epigenetic characteristics in somatic cells of Triatoma infestans (Klug). The review is organized well to discuss various aspects to propose the T. infestans as an attractive model for cellular research. Please address the following concerns during the revision to proceed further:

1) Please indicate the sources for data presented in Figures 1, 3, and 4 (similar to figure-2). Add proper citations and permission for reprint from the original publishers.

2)  Please describe the units 'C' used for describing ploidy degree in lines 61 and 67.

3) The order of figures citation in the text needs to follow the journal guidelines. Figure 2a/b were cited prior to Figure 1D in the text. Please correct and follow the guidelines.

4) Please try to provide critical methodological details to understand the data provided in the figures (adopted from other sources) by the readers.

5) Please add the full names of genes (such as KMT5C),and abbreviations such as DAPI (though standard ) to avoid confusion to the readers.

6) Though authors just emphasized the positives to propose T. infestans as a unique biological model, I would recommend to add few statements to indicate the limitations as well in the conclusions/summary.

Best,  

Minor edits are needed.

Author Response

Answers to questions:

  1. All figures in the revised version of the manuscript are exclusive and not previously published.
  2. The unit “C” was defined. Please see the revised text of the manuscript. Reference [9]
  3. The figure citation order was obeyed.
  4. Methodological details were provided in the legend of the figures.
  5. The lysine methyltransferase 5C and 4’,6-diamidino-2-phenylindole, dihydrochloride names were added in full.
  6. Limitations of these studies were introduced in the Conclusions and Abstract sections.

Thank you for your comments.

Reviewer 3 Report

The review by Maria Luiza Mello deals with the use of somatic cells of the insect Triatoma infestans as a model to study chromatin and nuclear organization.

According to the author, several relevant phenomena can be studied using those cells, such as somatic polyploidy and heterochromatin compartmentalization and its functional roles, expecially during differentiation. The paper is well written, the references are well chosen. This review is quite unprecedented, probably because the author is the only scientist investigating this model. It will be interesting for people interested in chromocenters, that are mostly investigated in drosophila and mouse cells.

The author mentions differences in the DNA content between two lineages. I was suprised that the consequences of those differences are not explored but probably the data are not available.

The legends of the various figures could be improved, especially for figure 1: it is not clear what the di shows and the p is absent. What is the heterochromatin unpacking that is supposed to be seen on 1d? Should figure 2 also present an untreated nucleus?

The "unexplained" word is probably not required in the abstract.

Author Response

Thank you for your comments!

The legends of the figures were improved, Fig. 2 was replaced by an exclusive and better one, and the “unexplained” word in the Abstract was deleted.

Regarding consequences of the differences in the DNA content between the two lineages I did not find available data.